# Regress, Don't Guess – A Regression-like Loss on Number Tokens for Language Models

**Jonas Zausinger**[1,2]   **Lars Pennig**[1,2]   **Kacper Chlodny**[1,2]   **Vincent Limbach**[1,2]

**Anna Ketteler**[1,2]      **Thorben Prein**[1,2]      **Vishwa Mohan Singh**[2,3]

**Michael Morris Danziger**[4]

**Jannis Born**[4,*]

[1]TU Munich, Germany; [2]TUM.AI, Germany; [3]LMU Munich, Germany
[4]IBM Research Europe, Switzerland;
Corresponding author: `jab@zurich.ibm.com`

## Abstract

While language models have exceptional capabilities at text generation, they lack a natural inductive bias for emitting numbers and thus struggle in tasks involving reasoning over quantities, especially arithmetics. This has particular relevance in scientific datasets where combinations of text and numerical data are abundant. One fundamental limitation is the nature of the CE loss, which assumes a nominal (categorical) scale and thus cannot convey proximity between generated number tokens. As a remedy, we here present two versions of a number token loss. The first is based on an $L_p$ loss between the ground truth token value and the weighted sum of the predicted class probabilities. The second loss minimizes the Wasserstein-1 distance between the distribution of the predicted output probabilities and the ground truth distribution. These regression-like losses can easily be added to any language model and extend the CE objective during training. We compare the proposed schemes on a mathematics dataset against existing tokenization, encoding, and decoding schemes for improving number representation in language models. Our results reveal a significant improvement in numerical accuracy when equipping a standard T5 model with the proposed loss schemes.

## 1   Introduction

As coined by Thawani et al. [14], numbers in natural texts are *ubiquitous* and *important*, yet systematically *neglected* by language models (LMs). Even worse, while Transformers [15] were invented for NLP, they have permeated various scientific domains (chemistry, biology, etc [2, 8, 1]), where tabular/numerical data is more prevalent than in NLP and often even fundamental for constructing task definitions: Molecules are labeled with drug efficacy, chemical reactions with yield, and synthesis procedures are natural text interspersed with quantities and times. Still, LMs notoriously struggle even with simple arithmetic tasks like three-digit multiplication [5] for multiple reasons:

1. **Tokenization**: Standard subword tokenization splits numbers into arbitrary tokens, disrupting their structure. Mitigation strategies include scientific notation [18] or digit-level tokenization [6], which may also preserve the decimal order of each digit [1].

MATH-AI Workshop @38th Conference on Neural Information Processing Systems (NeurIPS 2024).

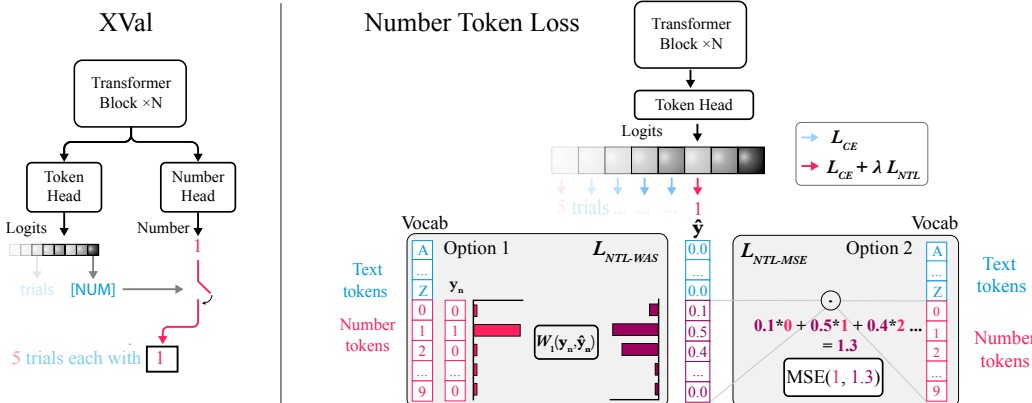

**Figure 1:** *Left*: xVal [7] decodes numbers through a regression head carried alongside the regular token head, gated through the [NUM] token (figure reproduced with permission). *Right*: Instead, the Number Token Loss (NTL) circumvents the need for two heads and allows the computation of a regression loss directly on the token head. We propose two schemes to achieve this: $\mathcal{L}_{\text{NTL-MSE}}$ (right) leverages a dot product of the values of the number tokens and their class probabilities. The $\mathcal{L}_{\text{NTL-WAS}}$ (left) uses the Wasserstein-1 distance of the (sorted) number token labels and their class probabilities.

2. **Embedding**: Canonically, the model has to recover the structure of numbers from data because the embeddings of numerical tokens are learned like any other token. Countless flavors of numeracy-preserving word embeddings exist [13, 1, 7], often akin to positional encodings.
3. **Training objective**: The standard cross-entropy (CE) loss assumes a nominal scale, thus it fails to convey the proximity between numbers, effectively inducing a semi-supervised setting. For example, predicting a [3] instead of a [2] token will not generally induce lower loss than a [9]. This problem has been surprisingly neglected and is the focus of this work.

Here, we aim to equip LMs with better inductive biases to handle combinations of textual and numerical data, such as math word problems or scientific datasets. In particular, we propose two versions of a regression loss on number tokens that respect numerical proximity (cf. Figure 1 right) and can be effectively combined with regular CE. The first version of this loss computes the Mean Squared Error (MSE) between the sum of the predicted class probabilities, weighted by their respective numerical token value, and the numerical token value of the label. The second version computes the Wasserstein distance between the distribution of the predicted number probabilities and the ground truth distribution, which is the one-hot encoding of the label. We integrate these improved training objectives with existing solutions for tokenization and embedding, in particular the Regression Transformer [1]. We evaluate all methods on a subset of the mathematical-question-answer dataset from DeepMind [12].

Prior art for joint language-number modeling suggested the use of verifiers [3, 10], calculators (typically: Python interpreters), or chain-of-thought (CoT) reasoning [19] to yield improved performance in Large Language Models (LLMs). We argue that all such strategies avoid the fundamental, underlying problem (i.e., number representation in LMs is poor) by reformulating the task, trying to correct answers *a posteriori* with calculators, or using significantly more compute (CoR). Therefore, we herein intentionally attempt to improve a classic, relatively small encoder-decoder LM with up to 220M parameters, namely T5 [11].

## 2 Methods

### 2.1 Number Token Loss

The idea of the Number Token Loss (NTL) is to add an additional loss term to the CE, which is only applied to number tokens and takes their numerical proximity into account. To achieve this, we propose two versions.

**Number Token Loss with Mean Squared Error (NTL-MSE)**    This loss compares the numerical value of the ground truth token with the weighted sum of the respective numerical token values, with the weights corresponding to the predicted class probabilities (cf. Figure 1 right). Given a model $f(\cdot)$, input tokens $\mathbf{x}_{\leq i}$ (where $i \leq N$), the numerical value $\hat{y}_i$ of ground truth token $y_i$ and a vocab $V$ consisting of tokens (with indices $j, ..., k$ representing the number tokens), we compute NTL-MSE:

$$\mathcal{L}_{\text{NTL-MSE}} = \frac{1}{N} \sum_{i}^{N} (\hat{y}_i - f(\mathbf{x}_{\leq i})_{j:k} \circ V_{j:k})^2 \tag{1}$$

Instead of a nominal-scale loss with regular CE, the NTL-MSE effectively conveys proximity between numbers. For example, if the label is [4] and the LM predicts a [5] instead of [9], the loss will be lower, matching our intuitive expectation, unlike the CE which gives constant loss no matter the proximity of the number (cf. Figure 2). This is sufficient for the vast majority of cases, however, since the NTL is not injective (like CE), it can return spuriously low loss for incorrect predictions. Consider e.g., a label [4] with $50\%$ of the mass on [0] and $50\%$ on [8] token, then NTL will be zero (Figure 3). While such cases are rare due to the softmax emphasizing logit differences, combining NTL with CE loss helps correct spurious cases, as CE continues refining predictions without reducing its value in these instances. However, to address this non-injectiveness, we propose a second version based on the Wasserstein-1 distance.

**Number Token Loss with Wasserstein-1 distance (NTL-WAS)**    This loss calculates the Wasserstein-1 distance between the predicted probability distribution of the (sorted) number tokens and the ground truth probability distribution, which is 1 for the label token and 0 for all other tokens. Given the ground truth $y_i$, a vocab $V$ with number tokens ordered from indices $j$ to $k$ and the cumulative distribution function $\text{CDF}(\cdot)$, we compute NTL-WAS:

$$\mathcal{L}_{\text{NTL-WAS}} = \frac{1}{N} \sum_{i=1}^{N} |\text{CDF}(y_i) - \text{CDF}(f(\mathbf{x}_{\leq i})_{j:k})| \tag{2}$$

As one can see in Figure 2, this version of the NTL not only conveys proximity between numbers correctly but also eliminates the non-injectiveness problem, shown in Figure 3. Both versions of the NTL are scaled with $\lambda$ (0.3 unless mentioned otherwise) and added to the regular CE loss:

$$\mathcal{L} = \mathcal{L}_{CE} + \lambda \mathcal{L}_{NTL} \tag{3}$$

Note that both versions of the NTL shall be 0 for all non-numerical tokens. By changing the $p$-order in NTL-MSE, different $L_p$-norm losses can be obtained (e.g., NTL-MAE). Huber loss is also compatible. In Appendix A.2, we provide pseudo-code for both versions of the NTL.

## 2.2    Backbone T5 and model variants

We use a T5 backbone [11] (Appendix A.3) for our experiments and extend it with both versions of the NTL and the Regression-Transformer tokenization scheme[1], due to its flexible encoder-decoder architecture and its success in various natural language processing tasks.

**Regression Transformer (RT).**    The Regression Transformer [1] tokenizes numbers on digit level, considering both the position and value of each digit. Since standard learnable embeddings may not adequately preserve the inherent structure of numbers, it leverages an inductive bias to account for the relative proximity of the numbers through numerical encodings, further explained in Appendix A.5.

**xVal encoding and decoding scheme.**    The xVal method [7] encodes real numbers using a single [NUM] token multiplied by its numerical value. For decoding (see Figure 1), a number head predicts the value while the token head outputs the sequence, replacing [NUM] during inference. However, this scheme is incompatible with T5 (see Appendix A.6). We thus use the xVal encoder and masked language modeling in our experiments.

**Integration of the Number Token Loss**    Both versions of our proposed NTL, depicted in the right panel of Figure 1, can be integrated into any model that treats numbers as clearly separated tokens of single digits by applying it as an additional loss term. Therefore, we adapt the tokenization scheme

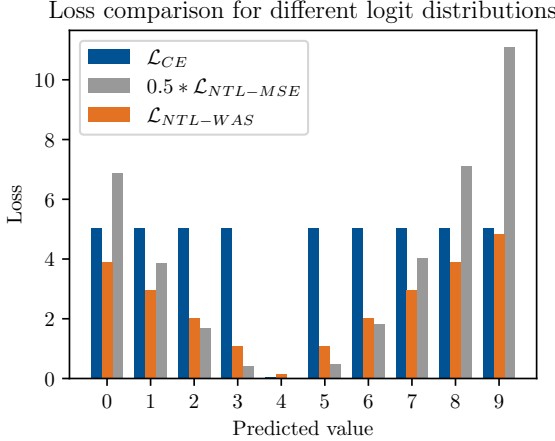

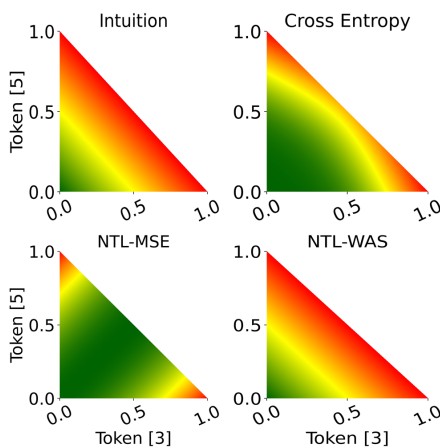

**Figure 2:** CE, NTL-MSE, and NTL-WAS for different predicted number values with ground truth label [4]. The underlying logit distribution over the simplified vocabulary (numbers 0-9) peaks at the respective predicted value and is uniform elsewhere.

**Figure 3:** The heatmap plot shows the respective loss for a given combination of the class probabilities for token 3 and 5, where the ground truth is token 4. The behavior of the NTL-WAS is closest to the intuitive desired behavior of the loss function, while the NTL-MSE does not have a unique minimum.

of the standard T5 model to tokenize all numbers on the digit level to make it compatible with the NTL. As RT already tokenizes numbers on digit level by default, we can integrate the NTL without any changes. Integrating NTL into xVal is not feasible, as xVal encodes every number with the same token. Moreover, xVal already uses both MSE and CE loss.

## 3 Experiments and results

To test the mathematical capabilities of the methods, we use a dataset with more than 25 million samples from the mathematical Q&A dataset from DeepMind [12]. The dataset comes with two sets of tests: interpolation tests, one for each type of question occurring in the training set, and extrapolation tests, which measure generalization along various axes of difficulty beyond that seen during training. We provide more information about the dataset in Appendix A.4. We evaluate all five models on the two test sets of this dataset and report the accuracy (how often the model predicts the number exactly), as well as the Mean Absolute Error (MAE) and the $R^2$-score. Since the dataset is skewed with some very high values, we perform a $log_{10}$ transformation on the predicted and ground truth numbers before calculating MAE and $R^2$-score.

All experiments except the one with xVal are built upon the T5 implementation and language modeling trainer based on the Hugging Face transformers library [16]. We use the T5-base model as a pretrained base for our respective models. All models were trained for approximately one million steps with a batch size of 32 over a period of approximately 3 days. More details on the models' training hyperparameters can be found in Appendix A.7.

**Table 1:** Evaluation metrics on test data.

(a) Interpolated Test Data

| Model | Acc. | MAE | $R^2$ |
|---|---|---|---|
| Standard T5 | .6448 | .1303 | .9688 |
| Standard + **NTL-MSE** | .7189 | .1091 | .9739 |
| Standard + **NTL-WAS** | **.7460** | **.0980** | **.9766** |
| RT | .7136 | .1135 | .9701 |
| RT + **NTL-MSE** | .6990 | .1291 | .9580 |
| xVal | .0000 | .2581 | .9735 |

(b) Extrapolated Test Data

| Model | Acc. | MAE | $R^2$ |
|---|---|---|---|
| Standard T5 | .3686 | 0.7847 | .9127 |
| Standard + **NTL-MSE** | .4278 | 0.7789 | .9091 |
| Standard + **NTL-WAS** | **.4324** | **0.7438** | **.9132** |
| RT | .4042 | 0.9868 | .7377 |
| RT + **NTL-MSE** | .4282 | 1.0988 | .6473 |
| xVal | .0000 | 0.8259 | .8186 |

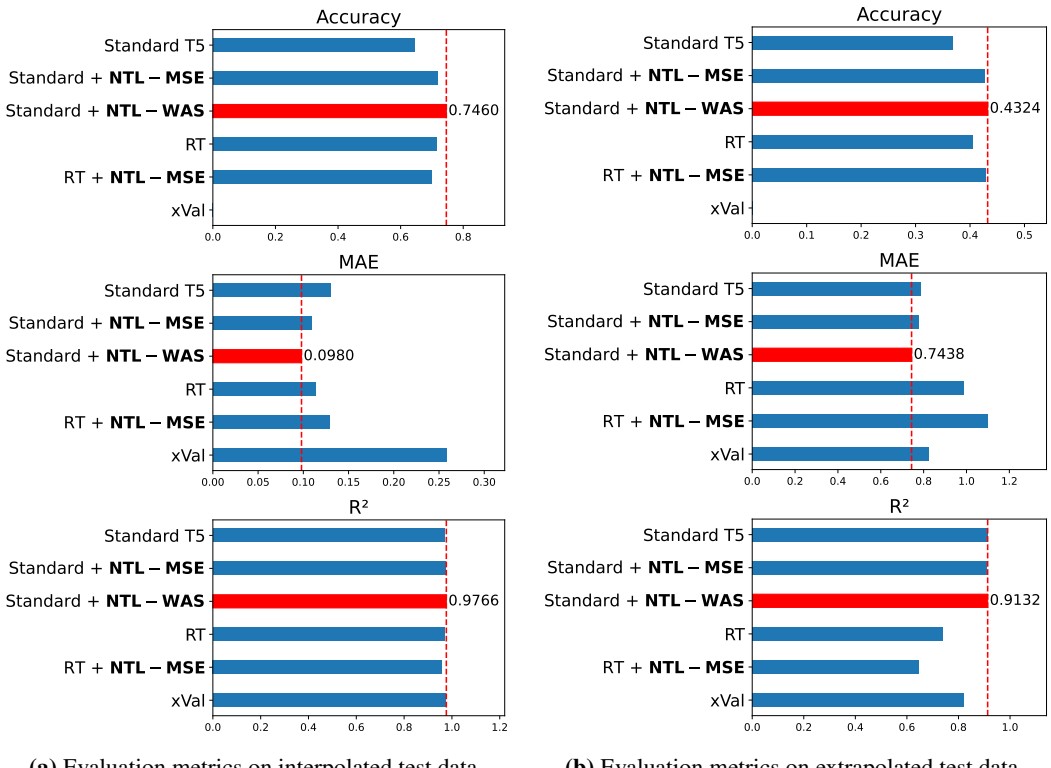

**(a)** Evaluation metrics on interpolated test data.

**(b)** Evaluation metrics on extrapolated test data.

**Figure 4:** Comparison of evaluation metrics on interpolated and extrapolated test data.

The results can be seen in Table 1 and Figure 4. They show that vanilla T5 clearly benefits from both our loss variants. Indeed, accuracy increases by more than $10\%$ for NTL-WAS compared to vanilla T5 in the interpolation tasks. The NTL-WAS was found to have the best performance across all three metrics and both interpolation and extrapolation tasks. This confirms our hypothesis that number representation in LMs can be effectively improved through a minor, architecture-agnostic modification of the loss function. The RT consistently surpasses vanilla T5 on interpolation, however no further benefit was found by augmenting RT tokenization with NTL-MSE, potentially due to the custom number embeddings conveying numerical proximity. The limited performance of xVal [7] is explained by the extensive range of numbers in the used dataset. The dynamic range of xVal is limited due to the combination of its scaling of the number token embeddings and the pre-layer-norm in the backbone [17]. As a result, the effective number range of xVal is limited to [-5, 5]. To take this into account, we scale our dataset for xVal with $log(1 + x)$. However, this means that large numbers can no longer be adequately distinguished by the model, as their embeddings become very similar.

## 4 Conclusion

We introduced the Number Token Loss (NTL) for LMs to enhance their ability to handle numerical data by considering the numerical proximity between tokens. Our experiments unambiguously demonstrate the effectiveness of the NTL-WAS loss. This confirms our hypothesis that number representation in LMs can be effectively improved through a minor, architecture-agnostic modification of the loss function. By augmenting the standard CE loss with NTL, we provide a simple yet effective method that integrates seamlessly into existing architectures without requiring additional computational overhead. Experiments on the DeepMind Mathematics Dataset demonstrated that NTL significantly improves numerical reasoning, especially in models without specialized numerical embeddings. This approach offers a practical solution for enhancing language models in numerically rich domains, paving the way for more accurate and reliable applications in mathematics and science.

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

# A Appendix

## A.1 Statement on code

The code for this paper is available at https://github.com/tum-ai/ibm_impact_project.

## A.2 Algorithm for the Number Token Loss

---

**Algorithm A1** Pseudo-code to compute NTL-MSE

---

1: **Initialize:** $\text{n\_vocab} \leftarrow \left[ \begin{cases} \text{int}(\text{vocab}[i]) & \text{if } \text{vocab}[i] \in \mathbb{R} \\ \text{NaN} & \text{otherwise} \end{cases} \right]_{i=1}^{V}$

2:
3: **function** FORWARD(logits $\in \mathbb{R}^{B \times T \times V}$, labels $\in \mathbb{R}^{B \times T}$) : Float
4:      ntl $\leftarrow 0$
5:      n_logits $\leftarrow$ logits$[:, :, \neg\text{n\_vocab.isnan}()]$               ▷ Ignore non-number tokens
6:      n_probs $\leftarrow$ Softmax(logits)
7:      $\hat{y} \leftarrow \sum_i \text{n\_probs}[:, :, i] \cdot \text{n\_vocab}$               ▷ $\hat{y}$ is $B \times T$
8:      $y \leftarrow \text{n\_vocab}[\text{labels}]$               ▷ $y$ is $B \times T$
9:      ntl $\leftarrow \text{MSE}(y, \hat{y})$               ▷ Can be any regression loss
10:      **return** ntl
11: **end function**

---

---

**Algorithm A2** Pseudo-code to compute NTL-WAS

---

1: **Initialize:** $\text{n\_vocab} \leftarrow \left[ \begin{cases} \text{int}(\text{vocab}[i]) & \text{if } \text{vocab}[i] \in \mathbb{R} \\ \text{NaN} & \text{otherwise} \end{cases} \right]_{i=1}^{V}$

2: **if** order_numbers is True **then**
3:      Sort the numbers in n_vocab by their numerical values
4: **end if**
5:
6: **function** FORWARD(logits $\in \mathbb{R}^{B \times T \times V}$, labels $\in \mathbb{N}^{B \times T}$) : Float
7:      n_logits $\leftarrow$ logits$[:, :, \neg\text{n\_vocab.isnan}()]$           ▷ Ignore non-number tokens
8:      n_probs $\leftarrow$ Softmax(logits)
9:      $y \leftarrow \text{n\_vocab}[\text{labels}]$            ▷ Retrieve true numerical values
10:      $\text{y\_distr}[b, t] \leftarrow \text{one\_hot}(y[b, t], \text{num\_classes}=\text{len}(\text{n\_vocab}))$      ▷ One hot encode y
11:
12:      $\text{wasserstein\_distance}[b, t] = \sum_{v=1}^{V} |\text{CDF}(\text{n\_probs}[b, t])[v] - \text{CDF}(\text{y\_distr}[b, t])[v]|$
13:
14:      ntl $\leftarrow \text{Mean}(\text{wasserstein\_distance}[\neg y.\text{isnan}()])$
15:      **return** ntl
16: **end function**

---

## A.3 T5 architecture

The T5 model is built upon the Transformer architecture [15], consisting of stacked self-attention and feed-forward layers in both the encoder and decoder. The encoder processes the input tokens to create contextualized representations, while the decoder generates the output tokens autoregressively, attending to both the encoder's outputs and the previously generated tokens. The model can be trained with both Masked Language Modelling (MLM) [9] and Causal/Auto-Regressive Language Modelling (CLM) [4], whereby we chose to use CLM.

## A.4 Dataset

To test the mathematical capabilities of the methods, we use a subset of the mathematical question-answer dataset from DeepMind [12]. The dataset was generated synthetically and therefore contains

limited linguistic variability, but is sufficient for our purposes to compare the mathematical capabilities of the different methods.

The dataset contains different modules and difficulty levels. For training and testing the models, we chose all difficulty levels but excluded modules where the answer contains complex fractions or variables. This allows us to focus on purely numeric answers to simplify the evaluation of the model and still leaves us with a large enough dataset of ~26 million samples.

For training, validation, and interpolation tests, we selected the following modules from the DeepMind mathematical question-answer dataset:

- `algebra__linear_1d.txt`
- `algebra__linear_1d_composed.txt`
- `algebra__linear_2d.txt`
- `algebra__linear_2d_composed.txt`
- `algebra__sequence_next_term.txt`
- `arithmetic__add_or_sub.txt`
- `arithmetic__add_sub_multiple.txt`
- `arithmetic__mul.txt`
- `numbers__div_remainder.txt`
- `numbers__div_remainder_composed.txt`
- `numbers__place_value.txt`
- `numbers__round_number.txt`
- `numbers__round_number_composed.txt`

For extrapolation tests, we selected the following modules:

- `arithmetic__add_or_sub_big.txt`
- `arithmetic__add_sub_multiple_longer.txt`
- `arithmetic__mixed_longer.txt`
- `arithmetic__mul_big.txt`
- `arithmetic__mul_div_multiple_longer.txt`
- `numbers__place_value_big.txt`
- `numbers__round_number_big.txt`

This resulted in a training dataset of 25,986,948 samples, a validation dataset of 13,026 samples, an interpolation test set of 130,000 samples, and an extrapolation test set of 70,000 samples.

## A.5 Regression Transformer

The Regression Transformer [1] preserves the inherent structure of numbers by inducing information on relative proximity through numerical encodings that are set deterministically for all tokens. For every combination of a decimal place and digit value, a corresponding numerical token is added to the vocabulary. For instance, the number $11.4$ is tokenized to `[1_1, 1_0, 4_-1]`.

Non-numeric tokens are set to zero vectors. The numerical encodings are designed so that their pairwise distances are symmetric and monotonically decreasing with the float value. The final encoding of the input tokens is obtained by summing over numerical and regular word encodings. The Regression Transformer numerical encodings NE at dimension $j$ for numerical token $t_{v,p}$ with value $v$ and decimal place $p$ can be determined by

$$\text{NE}_{\text{Float}}(v, p, j) = (-1)^j \cdot \frac{v \cdot 10^p}{j + 1}. \tag{4}$$

## A.6 Challenges with Integrating xVal in Transformer Models like T5

In transformer models like T5, integrating numerical encoding schemes like xVal presents challenges. Relative positional encodings and pre-layer normalization disrupt the numerical scaling. This makes it difficult to preserve distinctions between values.

In T5, instead of using absolute positions for each token, relative positions between tokens are encoded. This helps the model understand relationships between tokens based on their distance, regardless of where they appear in the sequence. However, this relative encoding is applied uniformly across all tokens, including numerical tokens. Since relative position encoding doesn't account for the magnitude of numerical values, it essentially ignores the scaling factor introduced by the xVal method.

Pre-layer normalization is applied to the inputs before they enter each transformer layer. Normalization typically scales the inputs to a standard range, effectively reducing the impact of the differences in numerical embeddings introduced by the xVal method. As a result, even though the xVal method multiplies the [NUM] token embedding by its corresponding numerical value, this scaling gets neutralized by the normalization step, making the embeddings of different numbers more similar than they should be.

## A.7 Training hyperparameters

We train each model for 1050000 iterations with a batch size of 32 using transformers[16] 4.42.4. We train with a learning rate of 1e-4 and weight decay of 0.01. All models were trained on single graphics processing units (GPUs) (NVIDIA RTX A6000). For the Number Token Loss, we trained with the hyperparameter $\lambda$ set to 0.3.

