# OpenReview forum: "Regress, Don’t Guess – A Regression-like Loss on Number Tokens for Language Models"
_NeurIPS.cc/2024/Workshop/MATH-AI — MATH-AI 24_

### Official Review · Reviewer_bzWf · 2024-10-05
**Promising Approach with Practical Applications, but Needs Improved Formatting and Justification of Results**

**Rating:** 6
**Confidence:** 4

**Review:**

### Summary and Final Decision
**Rating: 6 (Marginally above acceptance threshold)**

This paper introduces Number Token Loss (NTL), a novel method aimed at improving numerical reasoning in language models by addressing limitations in cross-entropy loss for handling numerical tokens. The authors demonstrate that NTL improves performance, particularly in interpolation tasks, and the method is easy to integrate into existing models, making it highly practical for real-world applications like finance and scientific data processing. However, improvements in formatting (e.g., figure placement and reducing excessive baseline explanations) and including variance in the results to justify the performance gains could push this paper to a 7 or 8.


### Strengths:

1. **Simple Idea**: The introduction of the **Number Token Loss (NTL)** is an elegant and simple solution to a complex problem—handling numerical tokens more effectively in language models.

2. **Clear Objective**: The paper has a focused goal—improving numerical reasoning in LMs—which is well-defined from the outset and directly addressed through the proposed methodology.

3. **Significant Results**: The experimental outcomes clearly show that NTL improves arithmetic reasoning, particularly in tasks involving numerical interpolation. This demonstrates the practical benefits of the proposed method.

4. **Practical Application**: The method is easy to implement and integrates seamlessly into existing architectures. This makes it highly practical for real-world applications, particularly in domains where numerical accuracy is crucial (e.g., scientific and financial data processing).

---

### Areas of Improvement:

1. **Formatting**: **Figure 1** starts the paper abruptly, making the flow feel disjointed. Additionally, **Figures 2 and 3** take up too much space, and their contribution is minimal beyond explaining the non-injectiveness of NTL (i.e., cases where the predicted average can match the target despite errors), which is already well covered in the text.

2. **Failure Mode of NTL**: The authors claim that the **non-injectiveness** of NTL is rare, but this issue may be more prevalent, especially in larger datasets. However an angle which is not explored is that the combination of **NTL and cross-entropy** can help with early convergence, and cross-entropy can address spurious cases as the model refines its predictions. This balance should be discussed more explicitly.

3. **Lack of Edge Case Exploration**: The paper could explore more **edge cases**, such as predictions where token-level accuracy is close, but the numeric difference is large (e.g., predicting 129 instead of 123 or 133). In such cases, NTL may perform worse than cross-entropy. Expanding this discussion would provide a fuller understanding of NTL's limitations.

4. **Excessive Explanation of Baselines**: The detailed descriptions of methods like T5 and xVal take up too much valuable space. A brief summary with references would suffice, freeing up room for a deeper exploration of the novel contributions.



### Criteria Ratings:

1. **Relevance to Journal**: 5/5
   - The paper aligns perfectly with the journal's focus on improving mathematical reasoning in LMs, particularly in the context of arithmetic and numerical tasks, which is highly relevant.

2. **Clarity of Language and Structure**: 3/5
   - While the paper's objectives are clear, the structure suffers from some abrupt transitions and excessive details on baseline models. The formatting of figures and the placement of certain explanations could be improved for better readability.

3. **Experimentation Procedure**: 3/5
   - The experimental setup is well-structured, but the results could be clearer. The performance differences are minimal, and without averaging across multiple runs, it’s difficult to determine if the improvements are real or just noise. A more thorough analysis would help distinguish meaningful gains, and **knowing the variance** in these results would also be beneficial to understand the consistency of the improvements.

4. **Innovation and Original Contribution**: 4/5
   - The introduction of the **Number Token Loss (NTL)** is a novel and impactful contribution. However, the lack of exploration of edge cases and the challenges with non-injectiveness leaves some room for further innovation.

5. **Reproducibility**: 4/5
   - The paper provides sufficient detail on model configurations, datasets, and hyperparameters, supporting solid reproducibility.




### Line-by-Line Review:

---

**Line 15 (Introduction)**:
*Figure 1 is placed immediately after the introduction.*

**Comment**: The placement of **Figure 1** disrupts the flow of the introduction, starting the paper with a figure rather than guiding the reader through key concepts. This figure should be moved to a later section where the methodology is discussed, allowing the introduction to flow more naturally and provide proper context before jumping into figures.

---

**Lines 47**:
*"We herein intentionally follow a simplistic scheme of attempting to improve a classic, small encoder-decoder LM, namely T5 [11]."*

**Comment**: Instead of using "small model," it would be clearer to specify the actual size of the model (e.g., "a 220 million parameter model"). This adds precision and avoids ambiguity, as "small" is relative in the context of language models.

---

**Line 51**:
*"We compute the weighted sum of the class probabilities with the numerical value of each token."*

**Comment**: Use the term **"predictions"** instead of **"values"** to distinguish between the model's outputs and the ground truth labels. This will clarify the difference between the predicted and true values in the context of NTL.

---

**Line 53 (Equation 1)**:
**Comment**: Be more specific that **"V"** refers to the **numerical tokens in the vocabulary** rather than embeddings. Clarifying this point will make it clear that the operation is being done on numeric tokens rather than embedded representations.

---

**Lines 59-60**:
*"Even though such cases are empirically rare because the softmax emphasizes differences in the logits, in practice, we always combine NTL with a regular CE loss."*

**Comment**: Rephrase to acknowledge that such cases may be more frequent than implied. Consider:
"While the authors claim these cases are rare, they may occur more frequently in practice. Combining NTL with cross-entropy loss helps mitigate this issue. NTL may accelerate early convergence by providing extra loss signals, but spurious cases (e.g., predicting an average value of 4 when the label is 4) can be corrected as the cross-entropy loss continues to refine predictions, since its value will not be reduced in such instances."

---

**Figures 2 and 3**:
*Line 63 - Discussion of Figures 2 and 3*

- **Figure 2**:
  **Comment**: This figure takes up too much space for a relatively small point that has already been explained in the text. It visually reinforces an obvious concept—the limitations of non-injectiveness in NTL—but doesn’t provide additional value. Reducing its size or removing it would improve the paper's readability.

- **Figure 3**:
  **Comment**: This is the third reiteration of the same concept, and together with Figure 2, takes up about a huge chunk of the paper's space. While it's important to clarify NTL’s non-injectiveness issue, this level of detail is excessive. Compressing the discussion and illustrations of this concept would allow more room for novel contributions and further analysis.


---

**Lines 67-71**:
*"For every combination of a decimal place and digit value, a corresponding numerical token is added to the vocabulary. For instance, the number 11.4 is tokenized to [1_1, 1_0, 4_-1]."*

**Comment**: The explanation of the Regression Transformer’s tokenization scheme is clear but takes up unnecessary space. Since this is a baseline method, a concise description and reference to the original work would be more appropriate. This would free up space for a deeper discussion of the novel contributions of the paper.  Alternatively consider moving this into a separate Related Work section.

---

**Lines 78-89**:
*"The xVal method [7] introduces a specialized numerical encoding and decoding scheme that represents any real number using just a single [NUM] token..."*

**Comment**: The explanation of **xVal** is detailed but overly long. A more concise summary, combined with a reference to the original work, would allow more space for analysis of the limitations of xVal in this specific context, particularly concerning scaling issues in the T5 architecture. Alternatively consider moving this into a separate Related Work section.

---

**Line 110 (Table 1)**:
*Performance metrics with accuracy, MAE, R2 scores.*

**Comment**: The differences in performance between models are minimal, making it difficult to discern meaningful improvements. Including variance or standard deviation in the results could strengthen the justification for improvements and demonstrate whether these improvements are consistent and significant or merely due to noise.

---

**Line 123 (Conclusion Section)**:
*"This approach offers a practical solution for enhancing language models in numerically rich domains."*

**Comment**: The conclusion could be enhanced by discussing future work. For instance:
"A deeper exploration of edge cases where token-level predictions are close but numerically different (e.g., predicting 129 vs. 123 or 133) would help clarify the limitations of NTL. Moreover, methods like the Regression Transformer (RT), which already handle numerical proximity well, could be integrated with NTL. Future work could explore combining RT’s strengths with NTL, potentially improving performance beyond what either method achieves alone."

---

### Official Review · Reviewer_xcgR · 2024-10-07
**The paper introduces Number Token Loss as an augmentation to Cross Entropy Loss to enable the feasibility of capturing number proximity relationship. The paper evaluates the method on T5 model and shows reasonable improvements**

**Rating:** 7
**Confidence:** 5

**Review:**

*Quality*

The paper uses deepmind math dataset and T5 open source model. The evaluation is limited to methods on T5 rather than a comparative analysis across llm and also the metrics MAE, R2 and Acc might not be sufficient to cover all the different reasoning tasks in deepmind math dataset. Despite the limitation I still think the current evaluation is still very helpful to improvise and explore more in this direction so not penalizing this limitation

Though the model and dataset are open source and replicating Number Loss Token is feasible , reproducibility is dependent on the sampling of dataset

*Clarity*

The paper is well written ,organized , structured and delightful to read.

*Originality*

Biggest missing piece was Related work. Though the number token loss mentioned in the paper is a new approach , there are various methods in which custom loss functions for numerical representation might have been tried before

*Significance of this work*

This paper has very good learning and novelty to proceed in the future directions. Adapting the method in other LLM based models and evaluating would be more fruitful. Also its important to address issues at token level rather than post training like mentioned in the paper and this work would help give the community more insights and encouraging results to research further

*Pros*

Reproducible with certain caveats

Addresses a significant problem with LLM

*Cons*

Dataset Sample limitation

Compute efficiency claim across any LLM is questionable

Not evaluated on other LLM models like LLAMA to generalize if NTL might work

---

### Official Review · Reviewer_TaEd · 2024-10-07
**Review on Augmenting Language Models with Regression Loss on Number Tokens for Arithmetic Reasoning**

**Rating:** 6
**Confidence:** 3

**Review:**

The study describes an important step in overcoming the well-known vulnerabilities of language models in processing numbers. Number Token Loss proposes such a task as well by introducing a loss function based on the distance between numerical predictions which may improve on tasks that require arithmetic reasoning but where the standard cross-entropy loss has proven ineffective. The experimental results obtained high evaluation scores on the DeepMind Mathematics Dataset leveraging better numerical performance. Nevertheless, the paper is in need of important modifications particularly with respect to the methodology and the technical aspects of the work. There are some shortcomings in this respect: the issues of NTL usage in various encoding-decoding architectures – Regression Transformer, xVal, etc. – are addressed too briefly. In addition, a broader experiment on other databases with a more thorough investigation of the benefits and weaknesses of the proposed solution should be conducted to support the paper arguments. However, enough of these concerns were raised and the main concept is still under-researched; with more workload it is conceivable that this work will be of great importance by its results.

---

### Official Review · Reviewer_aRcP · 2024-10-07
**Novel Loss Function for Numerical Data**

**Rating:** 7
**Confidence:** 3

**Review:**

This paper introduces a new loss function that works specifically on numerical tokens while ignoring linguistic tokens. It is then combined with other standard loss functions to compute the total loss. This is evaluated on a question-answer mathematics dataset. The results show a clear benefit when added to a T5 model. This new loss function is unique and addresses various issues in LMs when dealing with numerical data. There is a clear benefit in that it addresses proximity between numbers not considered in other loss functions.

It would have been nice to see more experiments done on other datasets. For example, the MATH dataset (https://openreview.net/forum?id=7Bywt2mQsCe) is quite common in this field, and assessing this new loss function on this dataset would be interesting as well. I would also like to understand more thoroughly the downside to adding NTL to the regression transformer. MAE actually increases when NTL is added, why do you think that is? I understand in line 113 the explanation for why RT does not benefit from NTL, but it seems to actively be getting worse on both datasets.

Overall, a very interesting paper and this shows a clear benefit of how LMs can process numerical data more efficiently.

---

### Decision · Program_Chairs · 2024-10-07

**Decision:**

Accept

**Comment:**

The reviewers agree that this paper presents a valuable contribution for the MATH-AI community.